# A Molecular Analysis of the Aminopeptidase P-Related Domain of PID-5 from *Caenorhabditis elegans*

**DOI:** 10.3390/biom13071132

**Published:** 2023-07-14

**Authors:** Anna C. Lloyd, Kyle S. Gregory, R. Elwyn Isaac, K. Ravi Acharya

**Affiliations:** 1Department of Life Sciences, University of Bath, Claverton Down, Bath BA2 7AY, UK; acl50@bath.ac.uk (A.C.L.); kg540@bath.ac.uk (K.S.G.); 2School of Biology, University of Leeds, Leeds LS2 9JT, UK; r.e.isaac@leeds.ac.uk

**Keywords:** aminopeptidase P, zinc metalloprotease, alphafold2, bioinformatics, dimerisation, PID-5, RNA-induced epigenetic silencing, *Caenorhabditis elegans*

## Abstract

A novel protein, PID-5, has been shown to be a requirement for germline immortality and has recently been implicated in RNA-induced epigenetic silencing in the *Caenorhabditis elegans* embryo. Importantly, it has been shown to contain both an eTudor and aminopeptidase P-related domain. However, the silencing mechanism has not yet been fully characterised. In this study, bioinformatic tools were used to compare pre-existing aminopeptidase P molecular structures to the AlphaFold2-predicted aminopeptidase P-related domain of PID-5 (PID-5 APP-RD). Structural homology, metal composition, inhibitor-bonding interactions, and the potential for dimerisation were critically assessed through computational techniques, including structural superimposition and protein-ligand docking. Results from this research suggest that the metallopeptidase-like domain shares high structural homology with known aminopeptidase P enzymes and possesses the canonical ‘*pita-bread fold*’. However, the absence of conserved metal-coordinating residues indicates that only a single Zn^2+^ may be bound at the active site. The PID-5 APP-RD may form transient interactions with a known aminopeptidase P inhibitor and may therefore recognise substrates in a comparable way to the known structures. However, loss of key catalytic residues suggests the domain will be inactive. Further evidence suggests that heterodimerisation with *C. elegans* aminopeptidase P is feasible and therefore PID-5 is predicted to regulate proteolytic cleavage in the silencing pathway. PID-5 may interact with PID-2 to bring aminopeptidase P activity to the Z-granule, where it could influence WAGO-4 activity to ensure the balanced production of 22G-RNA signals for transgenerational silencing. Targeted experiments into APPs implicated in malaria and cancer are required in order to build upon the biological and therapeutic significance of this research.

## 1. Introduction

Germ cells are required to transmit genetic information from generation to generation. Maintaining a high degree of germ-cell genome stability is therefore integral to the preservation of genetic material [1]. Exogenous and endogenous sources of DNA damage, including the movement of transposable elements (TEs), pose a constant challenge to germ-cell integrity. TEs are abundant, self-propagating DNA sequences that can be inserted into new locations within the genome [2]. Although TEs are evolutionary drivers, they pose an intrinsic threat to faithful genetic information transmission, where excessive damage results in germline abnormalities and ultimately organism sterility [3]. 

Many organisms have evolved protective mechanisms against TE activity, including an RNA silencing mechanism termed the Piwi/piRNA (Piwi-interacting RNA) pathway. When this pathway is lost in *Caenorhabditis elegans*, sterility is not immediately visible, with only a limited set of TEs reactivated [4]. This unique effect can be attributed to the action of the worm-specific argonaute proteins (WAGOs), which act in an additional small RNA silencing process termed RNA-induced epigenetic gene silencing (RNAe), providing multigenerational embryonic silencing (Figure 1) [4].

Maternally provided piRNAs can initiate RNAe and maintain silencing in the absence of additional piRNAs [4]. Although the mechanisms of RNAe establishment and maintenance are unknown, recent findings from a “piRNA-induced silencing defective” (Pid) mutation screen have identified a protein, PID-2, as being involved in this process, as well as two PID-2-interacting proteins termed PID-4 and PID-5. It has been proposed that these proteins are integral to the effective inheritance of this long-term silencing. They also affect Z-granule homeostasis and the level of 22G RNAs [4]. Whilst both PID-4 and PID-5 contain eTudor domains, PID-5 possesses an additional domain related to aminopeptidase P (APP).

The APP family (EC 3.4.11.9) (cytosolic APP-1, cell surface APP-2 and mitochondrial APP-3) comprises conserved metalloproteases that catalyse the cleavage of N-terminal residues from peptide substrates with proline residue at the P1’ position [6]. The pyrrolidine ring of the proline side chain confers conformational rigidity and thus resistance to hydrolysis, with specific peptidases required to cleave the Xaa-Pro peptide bond. These enzymes are widely distributed in mammals, contributing to protein homeostasis and the prevention of replication-related genome instability [7]. They have also been identified as targets for developing novel antimalarial drugs as they promote parasitic survival [8].

The PID-5 structure was predicted using the machine learning-based algorithm, AlphaFold2, and the domain relating to APP (PID-5 APP-RD) is shown in Figure 2 (green colour) below.

There is evidence that the PID-5 APP-RD may be a product of a recent segmental gene duplication event in which the region of DNA containing the *pid-4* and *app-1* genes has been repeated within the protein. This assessment is based on the adjacent location of these genes in the *C. elegans* genome (Figure 3). It is likely that a recombination event may have resulted in development of the *pid-5* gene, but its presence through evolution suggests an adaptive function has been achieved.

The biological role of PID-5 is not fully characterised, raising questions over APP-RD functionality. However, due to the role of PID-5 in gene regulation and germline immortality, it has been suggested that the presence of this domain may influence N-terminal proteolysis in RNAe. Placentino et al. [4] hypothesised that the additional domain may bind APP-1 substrates without N-terminal cleavage, or alternatively, heterodimerize with *C. elegans* APP-1. Heterodimerisation could provide APP-1 activity in granules containing PID-5 or prevent APP-1 homodimerisation that may be inhibitory if *C. elegans* APP-1 dimerisation is critical for activity.

The three-dimensional structure of cytosolic APP (APP-1) has been determined previously by the application of X-ray crystallography to four organisms, namely *Caenorhabditis elegans*, *Homo sapiens*, *Plasmodium falciparum* and *Escherichia coli*. A detailed analysis of these known structures has provided insight into the possible function of APP-RD in PID-5, as they share a common catalytic mechanism. Performing a comparison of these structures using different bioinformatics techniques and undertaking a critical analysis of the available data enabled us to perform functional annotation by analogy. These analyses provide further understanding of the potential role of the PID-5 APP-RD in RNAe.

To address the Placentino et al. [4] hypotheses, this study focuses on the following key areas: (1) The assessment of the homology between the PID-5 APP-RD predicted structure and the available APP-1 crystal structures. (2) A comparison of metal composition across the metalloproteases and the catalytic mechanism. (3) A comparison of interactions, with a known APP-1 inhibitor used to assess potential substrate binding. (4) The evaluation of potential dimerisation interface interactions to assess the viability of PID-5 APP-RD homodimerisation and heterodimerisation with *C. elegans* APP-1. 

## 2. Materials and Methods

### 2.1. PID-5 and APP-1 Structures and Homology

The predicted structure of PID-5 (Q9GUI6) was obtained from the AlphaFold2 Protein Structure Database 2.0 (https://alphafold.ebi.ac.uk (accessed on 7 February 2023)) [9]. APP-1 structures for *Caenorhabditis elegans* (*Ce*APP-1), cytosolic human XPNPEP1 (HuAPP-1), *Plasmodium falciparum* (*Pf*APP-1) and *Escherichia coli* (*Ec*APP-1) were obtained from the Protein Databank Bank (PDB). FASTA format sequences were obtained from the UniProtKB database (Release 2023_01) (https://www.uniprot.org/ (accessed on 7 February 2023)). Structural alignment using the *Coot* molecular graphics application (0.9.8.7) enabled the isolation of the C-terminal APP-RD of PID-5 (S452-I1061) for homology assessment [10]. 

Multiple sequence alignment (EMBL-EBI 1.3.4) of PID-5 APP-RD and its homologues was performed using Clustal Omega (https://www.ebi.ac.uk/Tools/msa/clustalo/ (accessed on 28 February 2023)) [11]. A phylogenetic tree was constructed following ClustalW alignment with MEGA11 [12]. Pairwise sequence alignment from EMBOSS Needle quantified the sequence conservation of the homologues relative to PID-5 APP-RD (https://www.ebi.ac.uk/Tools/psa/emboss_needle/ (accessed on 3 April 2023)). DALI pairwise structure comparison was used to determine the structural homology (http://ekhidna2.biocenter.helsinki.fi/dali/ (accessed on 18 February 2023)) [13]. These tools are discussed in more detail in the Section 3.

### 2.2. Metal-Coordinating Residues and Composition

The PID-5 APP-RD was superimposed by secondary-structure matching in *Coot* [10], with *Ce*APP-1 (PDB code: 4S2R), HuAPP-1 (PDB code: 3CTZ), *Pf*APP-1 (PDB code: 5JQK) and *Ec*APP-1 (PDB code: 1WL9), respectively, used to assess the conservation of metal-coordinating residues. HuAPP-1 and HuAPP-2 (AlphaFold2 DB: 043895) were also superimposed in order to analyse the differences between isoforms.

The AlphaFill online interface (https://alphafill.eu/ (accessed on 8 February 2023)) was used to predict potential PID-5 APP-RD cofactor interactions [14]. The AlphaFill output was used for all subsequent analyses. 

### 2.3. Apstatin Inhibitor Interactions

A docking simulation of the PID-5 APP-RD predicted structure with apstatin was performed using GOLD software (2022.3.0) [15]. The coordinates and geometry of apstatin were generated in Chem3D (21.0.0). Apstatin-bound APP-1 structures (*Ce*APP-1:apstatin [PDB code: 4S2T], *Pf*APP-1:apstatin [PDB code: 5JR6] and *Ec*APP-1:apstatin [PDB code: 1N51]) were used as reference structures. The potential apstatin binding-site residues were determined by the superimposition of PID-5 APP-RD with the known apstatin-bound APP-1 structures. The apstatin-bound and unbound structures of *Ce*APP-1 (*Ce*APP-1:apstatin [PDB code: 4S2T and *Ce*APP-1 [PDB code: 4S2R]) were compared to assess any conformational changes that occurred upon binding. Equivalent residues to those identified as dynamic in *Ce*APP-1 (Glu929 and Arg941) were defined as flexible. Residues Glu967 and His932 were marked as deprotonated to mimic physiological conditions. A total of 6 poses were generated by the ChemScore function.

### 2.4. Dimerisation Interface Interactions

The *Ce*APP-1 interface was assessed through PDBePISA (Protein Interfaces, Surfaces and Assemblies) (https://www.ebi.ac.uk/pdbe/pisa/ (accessed on 19 February 2023)) [16]. PID-5 APP-RD was mapped onto both *Ce*APP-1 chains in *Coot*. Equivalent residues were noted and potential heterodimer interface interactions with *Ce*APP-1 were assessed. PID-5 APP-RD was also modelled as a homodimer. The protein surface charge potential and hydrophobicity was calculated using CCP4mg [17]. The ColabFold (1.5.2-patch) ‘AlphaFold2 using MMseqs2’ software was used to predict the homo- and hetero-oligomeric protein structures using PDB100 template mode (https://github.com/sokrypton/ColabFold (accessed on 5 July 2023)) [18]. PDBePISA was used to calculate the potential solvent-accessible area, buried upon dimer formation, for the predicted structures. Figures were produced using CCP4mg [17] and BioRender.com (accessed on 6 March 2023).

## 3. Results

### 3.1. PID-5 APP-RD and APP-1 Share High Amino Acid Sequence and Structural Homology

A comprehensive analysis of the sequence and structure conservation between known APP-1 structures and PID-5 APP-RD was performed. High percentages of similarity are indicative of functional relationships and therefore may provide information on the role of PID-5. In the absence of an experimentally solved structure for PID-5, the AlphaFold2-predicted structure (Figure 2) was used for comparison to crystal structures of *Ce*APP-1, HuAPP-1, *Pf*APP-1 and *Ec*APP-1.

The confidence levels of the PID-5 model were analysed prior to undertaking structural interpretation. AlphaFold2 provides confidence estimates per residue, using a predicted local distance difference test (pLDDT) that uses the local distance difference test Cα (IDDT-Cα) to estimate the level of agreement of the predicted structure to known experimental structures flagged during the multiple sequence alignment (MSA) step of the AlphaFold2 algorithm [19]. The APP-RD of PID-5 returned scores in the ‘Confident’ to ‘Very high’ model confidence range (90 > pLDDT ≥ 70 and pLDDT ≥ 90, respectively), where scores above 70 indicate a good backbone prediction [20]. Several residues fell within the ‘Low’ confidence category (70 > pLDDT ≥ 50). The residues were Asn526, Ser587, Ser992-Gln996 and Glu950-Asn955 (Figure 2C, coloured yellow) and they were interpreted with caution in subsequent analyses.

An MSA was performed to assess the similarity of the PID-5 APP-RD to known APP-1 structures at the sequence level. Areas of high conservation across all structures were revealed, particularly within the putative active site of PID-5 APP-RD (residues 908–947). (Figure 4). We also see high conservation between residues 958–982 and 859–890. 

The MSA output was used to create a phylogenetic tree to assess the evolutionary relationships between the structures (Figure 5). *Ce*APP-1 was returned as the closest relative to PID-5 APP-RD. 

A pairwise sequence alignment was performed to quantify the percentage of perfectly conserved amino acids and those with conserved physiochemical properties in PID-5 APP-RD, relative to the homologous structures (Table 1). The *Ce*APP-1 sequence ranked the highest, with 41.4% shared identity with and 61.4% similarity to PID-5 APP-RD. 

Structural comparison can provide information beyond sequence comparison and provide additional insights into functional conservation [13]. A DALI structural search was performed to compare the PID-5 APP-RD predicted structure to the crystal structures of its homologues (Table 2). Each result obtained a Z-score > 20, which confirmed that PID-5 APP-RD is homologous to all analysed structures [22]. The search returned *Ce*APP-1 as the highest structural homologue, with a Z-score of 47.9.

Superposition of the two structures revealed the extent of the sequence conservation and 3D homology (Figure 6A), with PID-5 APP-RD returning the highest sequence and structural similarity to *Ce*APP-1. 

The ‘*pita-bread*’ fold aminopeptidases share the C-terminal peptide fold of two ααβββ repeats, which provides the structural foundation for catalysis [23]. The conservation of this fold was seen in the PID-5 APP-RD predicted structure (Figure 6B). As the fold is conserved, the substrate binding modes and catalytic mechanisms are also likely to be somewhat similar [24]. 

### 3.2. PID-5 APP-RD May Bind a Single Zinc Ion

The metalloaminopeptidase classification of APPs results from the presence of one or two divalent metal ions at the active site, which is central to the catalytic mechanism [25]. Metal ions can coordinate a water molecule that acts as a nucleophile for use in the hydrolysis of the peptide bond. All APP-1 members have dinuclear metal centres, with *Ce*APP-1 known to coordinate Zn^2+^ [26] and with HuAPP-1, *Pf*APP-1 and *Ec*APP-1 known to bind Mn^2+^ [27,28,29]. The two metal binding sites (M_A_ and M_B_) have differential affinities, with the M_A_ ion being more tightly bound [23]. A comparison of the conserved metal-coordinating residues from the homologous proteins with the PID-5 APP-RD predicted structure revealed the feasibility of metal ion binding in the PID-5 APP-RD (Table 3).

The spatial arrangement of the metal-coordinating residues was analysed to assess the potential for metal coordination in PID-5 APP-RD (Figure 7). 

On the basis of the amino acid sequence alignment and structural comparison, the conserved metal-ion-coordinating residues were defined as DEEH for M_A_ and DDE for M_B_. The interactions of the first glutamic acid side chain and the imidazole nitrogen atoms from the histidine in M_A_ were conserved in PID-5 APP-RD (E967 and H932). The other two metal-coordinating residues for M_A_ were identified as aspartic acid and glutamic acid. The equivalent residues in PID-5 APP-RD are asparagine (N868) and glycine (G981), respectively. Asparagine may form a coordinate covalent bond to stabilize the metal ion in the active site, but any interactions from the glutamic acid carboxylate group are lost. 

In the second metal-binding site (M_B_), the identity of the three coordinating residues was not conserved. Two negatively charged aspartic acid side chains in the homologues (Figure 7A–D) were shown to be equivalent to the uncharged asparagine and glutamine residues in PID-5 APP-RD (Figure 7E). The third coordinating residue was the M_A_ glutamic acid, which was equivalent to glycine in PID-5 APP-RD. Although a lone pair from the asparagine and glutamine carboxamide groups may interact with a metal through a coordinate covalent bond, the combined interactions from these mutated residues are likely to be inadequate to coordinate a metal ion at M_B_.

The analysis of the potential interactions from the mutated residues supports the hypothesis that PID-5 APP-RD could coordinate a single metal ion at M_A_ (Figure 7E). The inspection of the residues within the putative metal-binding site revealed both zinc and manganese ions to be biochemically plausible. 

Further investigation conducted using AlphaFill supports this hypothesis (Table 4). Where AlphaFold2-generated models lack ligand interactions and the presence of cofactors, AlphaFill can be used to consider these additional relationships [14]. Homologous structures are compared and any molecules and ions that have been experimentally shown to be present are assessed in the AlphaFill interface, resulting in an enhanced predicted model. AlphaFill returned five potential ions, three of which were consistently modelled as mononuclear metal centres at M_A_ (Zn, Mn, and Co), aligning with the single-coordinate hypothesis (Table 4).

The local RMSD for zinc from *Ce*APP-1 (PDB code: 4S2R) was > 0.64 Å and therefore was flagged as exhibiting medium confidence by AlphaFill. However, a low TCS was shown that was indicative of good transplant reliability. Manganese transplanted from HuAPP-1 (PDB code: 3CTZ) returned local RMSD and TCS scores in the high confidence bracket. The coordination geometry of the additional transplants was assessed. In the reference structures, the ions were either coordinated by more acidic residues, or were positioned away from the active site, and the cobalt transplant (PDB code: 1WN1) had lower coordinating residue homology than the manganese and zinc transplants.

To assess potential structural differences in manganese and zinc coordination, the structures of the human APP isoforms were studied. Interestingly, cytosolic HuAPP-1 binds manganese, whereas the membrane-bound HuAPP-2 has been characterised as zinc-binding [30]. The superposition of HuAPP-1 (PDB code: 3CTZ) onto HuAPP-2 (AlphaFold2DB: O43895) showed the complete amino acid identity of the metal-coordinating residues. Due to the amino acid conservation, the environment in which the protein resides may dictate which metal ion binds [30]. *Ce*APP-1 is a cytosolic Zn^2+^-binding protein and PID-5 is also cytosolic. Additionally, PID-5 is localised in the perinuclear region and in P granules [4]. Elemental tomography of *C. elegans* showed manganese to be predominantly localised to the intestine, whereas zinc was shown to be more widely distributed, and importantly, present in the gonads and the developing embryo [31]. 

Mononuclear zinc metalloproteases have a conserved active-site signature HExxH motif, where the two histidine residues contribute to the zinc coordination [32]. *Ce*APP-1 and HuAPP-2 have a HGTGH motif, whereas PID-5 APP-RD has the standard mononuclear Zn^2+^ metalloprotease motif (HETGH) between His928 and His932, satisfying the requirement for zinc binding. However, His928 is positioned ~8.95 Å (as measured from His928 NE2) away from the AlphaFill-positioned zinc, whereas the equivalent coordinating residue in *Ce*APP-1 (His496) is positioned ~4.5 Å from Zn^2+^. The rotation of Arg979 in PID-5 APP-RD from the current position in the predicted structure would allow His928 to move closer to the proposed Zn^2+^ ion. 

In conclusion, our analysis predicts that PID-5 APP-RD could coordinate one divalent metal atom at M_A_. The extent of homology with zinc-binding *Ce*APP-1, the distribution of Zn^2+^ compared to Mn^2+^ in *C. elegans* and the presence of the classical HExxH motif suggest that the M_A_ metal will be a Zn^2+^ ion. 

### 3.3. PID-5 APP-RD Could Bind the APP-1 Inhibitor Apstatin

The ‘*pita-bread’* aminopeptidases have analogous catalytic mechanisms that are highlighted by similarities in enzyme inhibitor binding. Apstatin is a non-hydrolysable peptide analogue that is known to be an APP-1 inhibitor through substrate mimicry (Figure 8) [24]. 

Apstatin-bound APP-1 structures are available for *Ce*APP-1 (PDB code: 4S2T, 2.15 Å), *Pf*APP-1 (PDB code: 5JR6, 2.3 Å) and *Ec*APP-1 (PDB code: 1N51, 2.3 Å). A total of 12 residues within the active site are conserved across the APP-1 domains analysed here, of which 5 are conserved in PID-5 APP-RD. In comparison to its most closely related homologue, *Ce*APP-1, PID-5 APP-RD shares 7 identical residues and has 11 ‘similar’ residues (Table 5). 

The 5 perfectly conserved apstatin-interacting residues may therefore be involved in apstatin binding for PID-5 APP-RD. To assess the potential interactions, the PID-5 APP-RD predicted structure was superimposed with the apstatin-bound crystal structures (Figure 9). His928 and Arg979 may contribute to the hydrophobic pocket occupied by the P1’ proline substrate (in homologous structures) binding and facilitate its correct orientation. Ile828 may contribute to the hydrophobic pocket occupied by the P1 phenylalanine. Residue Glu967 may function to activate the nucleophilic water molecule [34] and His932 may form a hydrogen bond, as seen in *Ce*APP-1:apstatin.

The conformational changes that occur upon binding were also considered by a comparison of the apo-*Ce*APP-1 (PDB code: 4S2R) and apstatin-bound *Ce*APP-1 structures (PDB code: 4S2T). In *Ce*APP-1, Arg505 flips 180° to undergo an interaction with apstatin, with no other conformational changes observed. Potential steric clashes were identified that could prevent apstatin from binding, especially due to minimal conformational change. The binding pose of *Ce*APP-1 apstatin with PID-5 APP-RD revealed four amino acid residues in close proximity to the ligand (Figure 9). To assess potential dynamics of PID-5 APP-RD and the potential binding residues, GOLD was used to dock apstatin into the putative binding site [15]. Arg941 was marked as flexible for docking due to the proximity to apstatin, enabling 34 rotamers to be assessed from the GOLD library. Glu929 had mutated from a glycine residue in *Ce*APP-1 and was therefore also marked as flexible, with 8 rotamers allowed. 

The docking returned 6 solutions, with the highest scoring solution displaying a binding pose in a similar orientation to *Ce*APP-1:apstatin for the P1 phenylalanine (Figure 10A). A total of 10 interactions were predicted (Figure 10B), although the potential hydrogen bond from Glu950 must be treated with caution due to ‘Low’ AlphaFold2 confidence. 

The positioning of docked apstatin further from the metal binding sites relative to *Ce*APP-1:apstatin could be attributed to the proposed absence of any interactions with M_B_. This was as apstatin coordinates both divalent metal ions in the homologous structures. 

The mutation of *Ce*APP-1 Arg78 and His392 into the less bulky Gly527 and Ala839 in PID-5 APP-RD, respectively, would provide a larger substrate binding pocket in PID-5 APP-RD. Indeed, the equivalent His392 (*Ce*APP-1)/Ala839 (PID-5 APP-RD) residue in *Pf*APP:apstatin (His551) was shown to provide an ‘acute bend’ between the S1 and S1’ pocket [28], and may explain the different orientation of the P1’ proline in the PID-5 APP-RD-docked apstatin model (Figure 10A).

The predicted hydrogen-bonding interactions were compared to the known crystal structures of *Ce*APP-1, *Pf*APP-1, and *Ec*APP-1 in a complex with apstatin, and it was shown that all the homologues utilise at least 6 interactions for apstatin binding (Appendix A). Although mutated residues (Table 5) meant that the other equivalent hydrogen-bonding interactions in the homologues were lost, a comparison of the docking results to the interactions present in *Ce*APP-1:apstatin suggests that some residues may form compensatory interactions (Appendix A). 

The PID-5 APP-RD docking results suggests that five amino acids could be used to form 8 hydrogen bonds with apstatin (Figure 10B), compared to the seven amino acids in *Ce*APP-1 that form 13 hydrogen bonds [26]. The occurrence of fewer potential hydrogen-bonding interactions may have also contributed to the difference in the docked position relative to *Ce*APP-1:apstatin. 

Successful GOLD docking suggests apstatin binding is plausible for PID-5 APP-RD, although the fewer number of potential interactions would result in weaker binding. 

### 3.4. PID-5 APP-RD Could Heterodimerize with CeAPP-1

The formation of stable macromolecular complexes can relate to physiological function. Both native *Ce*APP-1 and *Ce*APP-1:apstatin are dimeric [26]. HuAPP-1 and *Pf*APP-1 also form dimers [27,28] and *Ec*APP-1 forms a tetramer (a dimer of dimers) [24]. The homodimeric structure may control access to the active site and contribute to the mechanism of action [28]. 

A PDBePISA-generated Δ^i^G (observed solvation free energy gain) *p* value of 0.176 suggested that the *Ce*APP-1 interface had high hydrophobicity compared to average structures and this was suggestive of a biological dimer. Previous research also showed detectable amounts of monomeric (2.1%) and tetrameric (9.6%) *Ce*APP-1 species following analytical ultracentrifugation [26]. PDBePISA was used to assess the dimer interface interactions of *Ce*APP-1, returning 16 hydrogen bonds and 4 salt bridges (Table 6).

Previous immunoprecipitation–mass spectrometry (IP-MS) experiments showed an enrichment of APP-1 with PID-5 that could be due to *Ce*APP-1:PID-5 heterodimerisation [4]. PID-5 APP-RD was superimposed onto a monomer of *Ce*APP-1 to assess the likelihood of PID-5 APP-RD heterodimerisation. Two copies of the PID-5 APP-RD predicted structure were also superimposed onto the individual *Ce*APP-1 monomers in order to model homodimerisation (Table 6).

Most of the interface interactions could be conserved, with the exception of two salt bridges, as *Ce*APP-1 Lys136 is equivalent to Gln583 in PID-5. However, hydrogen bonding-interactions may still form. The side chain interaction of *Ce*APP-1 Thr468 would also be lost as the equivalent residue is Ile913 in PID-5 APP-RD. However, the high residue conservation suggests that both the heterodimeric and homodimeric structures could satisfy the observation that, on average, 5-10 hydrogen bonds are seen per 1000 Å^2^ of protein interface [16].

*Ce*APP-1 and PID-5 APP-RD display a high conservation of charged residues at the dimer interface that may provide favourable electrostatic interactions between the monomers and contribute to the stability of a potential heterodimer (Figure 11). 

To further explore dimerisation, we predicted the structures of the heterodimer (Figure 12) and homodimer using AlphaFold2. The solvent-accessible area buried upon dimer formation was calculated by PDBePISA (Table 7) for the predicted structures [16].

The identification of potentially conserved hydrogen-bonding interactions and salt bridge formation, in addition to the demonstrated association between PID-5 and APP-1 [4], suggests that PID-5 APP-RD and *Ce*APP-1 heterodimerisation is feasible. A surface diagram of the AlphaFold2-predicted heterodimer highlights the large potential interface area (Figure 13). 

Previous research identified a tryptophan residue as vital for dimerisation in HuAPP-1, and when a Trp477 point mutation to glutamic acid blocked homodimerisation, only 6% of the wild-type activity was maintained [27]. This residue is conserved in *Ce*APP-1 (W475) and PID-5 (W920), and all three residues are positioned in the same locality and orientation. As this tryptophan residue is essential in the second closest homologue to PID-5 APP-RD and is present in both *Ce*APP-1 and PID-5, Trp920 can perform the role of *Ce*APP-1 Trp475 in the heterodimer which may help to facilitate dimerisation and maintain APP-1 activity. However, it is important to note that *Pf*APP-1 and *Ec*APP-1 do not have the tryptophan residue and still form dimers. 

Although the evidence suggests heterodimerisation is possible, the effect of such interactions remain unclear as the relationship between dimerisation and *Ce*APP-1 activity is currently unknown. Additionally, the potential substrates for the *Ce*APP-1:PID-5 APP-RD heterodimer are unconfirmed, although WAGO-4 has been identified as a potential substrate [4].

## 4. Discussion

### 4.1. AlphaFold2 and APP-1 Enabled Homology-Based Annotations

The AlphaFold2 structure prediction enabled the analysis of the PID-5 APP-RD in the absence of experimental data. Due to the high level of sequence and structural similarity of PID-5 with known APP-1s, strong analogies could be drawn between the structures, allowing us to infer the characteristics of PID-5 through association. The assumption that the predicted model is representative of the crystal structure underlies this study, although it should be acknowledged that AlphaFold2 has been recognised for its predictive accuracy and that the per-residue reliability estimates provided confidence in the analysis. 

### 4.2. Mononuclear Zinc Binding PID-5 APP-RD Is Probably Catalytic Inactive

This comparative structural study suggests PID-5 APP-RD may bind a single Zn^2+^ at M_A_, but it was not possible to consider the stability of the potential coordination complex in this report. The distance of the first histidine residue of the HExxH motif (His928) may be too great to facilitate a stable Zn^2+^ coordination and non-specific metal binding may occur as a result. As PID-5 APP-RD may also bind Mn^2+^, it may sense the concentration of different ions, with metal selectivity potentially having a role in modulating its activity. Indeed, the metal ion content of both HuAPP-1 and *Ec*APP-1 has been shown to vary under different cell culture conditions [27,35]. Furthermore, it has been reported that the mutation of the conserved metal-coordinating residues Asp260 and Asp271 in *Ec*APP-1 results in a catalytically inactive enzyme [34]. Additionally, the mutation of the active site residue His243 into alanine in *Ec*APP-1 abolished any catalytic activity [34]. The equivalent residue to His243 in PID-5 APP-RD via structural superimposition is Ala839. This evidence suggests mononuclear zinc PID-5 APP-RD will be catalytically inactive, and perhaps could be considered a ‘*pseudo-APP*’ molecule. Experimental evidence is required to confirm if a single metal does coordinate. 

### 4.3. Apstatin Binding Is Likely to Be Transient in PID-5 APP-RD

An IC_50_ value of 20.2 ± 1.2 μM was shown for *Pf*APP-1 [28], but apstatin is a relatively poor APP-1 inhibitor. This is particularly of its capacity to inhibit HuAPP-1, where a crystal structure of the protein–ligand complex is unavailable. This study suggests that apstatin binding to PID-5 APP-RD is more transient than in the homologous structures and therefore would be a very weak interaction. However, in vitro studies are required to confirm this. A peptide-PID-5 APP-RD complex may not be stable and, based on the current evidence, *Ce*APP-1 would outcompete PID-5 APP-RD for substrate. PID-5 APP-RD may therefore not function to bind and lock substrates without cleavage, as was suggested in the first Placentino et al. hypothesis [4]. An alternative role may be more probable, supporting the heterodimerisation hypothesis. 

### 4.4. Would Heterodimerisation Result in APP-1 Inactivity?

Placentino et al. [4] identified APP-1 as an interactor of PID-5 and this study showed heterodimer formation is plausible based on the high level of sequence conservation at the dimer interface. Placentino et al. [4] hypothesised that heterodimerisation may function to prevent APP-1 homodimerisation and activity, or to bring APP-1 activity into PID-5-positive locations. Although the relationship between dimerisation and APP-1 activity is unknown, the structural homology of *Ce*APP-1 and PID-5 APP-RD, particularly at the dimer interface, indicates the possibility that the heterodimer can maintain the correct fold for activity. 

Confirming the relationship between dimerisation and APP-1 activity would help to determine if heterodimerisation, with a likely catalytically inactive PID-5, results in APP-1 activity and hence enable the two hypotheses to be distinguished. 

## 5. Biological Implications

PID-5 is known to interact with PID-2, with a proposed role in regulating RdRP and other factors that bring about heritable silencing [4]. PID-4 has also shown to be a PID-2-interactor and both PID-4 and PID-5 contain eTudor domains. A lack of the characteristic aromatic cage and acidic amino acids suggests that they will not bind symmetrically di-methylated arginines and can use these domains to bind PID-2 non-simultaneously. Tudor domains are known to function as ‘adaptors’ [36], and therefore the role of PID-4/5 may be to act as mediators of the downstream effectors PMRT-5 and APP-1, respectively, for the modification of RNAe proteins.

This study shows that the heterodimerisation of APP-1 with PID-5 is plausible, connecting N-terminal proteolysis to RNAe. Whilst PID-4 may bring PMRT-5 activity to PID-2 for potential arginine modifications, PID-5 could bring APP-1 activity to the Z-granules and act on substrates such as WAGO-4 [4]. This argonaute protein interacts with 22G-RNAs and is a known promoter of heritable epigenetic silencing with ZNFX-1 [37]. It has been suggested that ZNFX-1 maintains epigenetic signals at the 3’ region of the target mRNA by redistributing RdRPs, as argonaute proteins tend to target the 5’-end of mRNA [38]. ZNFX-1 and WAGO-4 may therefore work together to maintain a balanced 22G-RNA population and an even distribution of epigenetic signals [39]. WAGO-4 has a proline-rich (22.9% of the first 70 residues) N-terminal domain that comprises three proline dipeptides in the first 25 residues. Since two adjacent prolines of apstatin and bradykinin, a 9 amino acid peptide substrate, are well accommodated by the S1’ and S2’ subsites of *Ce*APP-1 [40], it is conceivable that the dipeptide motifs of WAGO-4 may be important for any interaction with the PID-5 APP-RD domain.

Placentino et al. [4] showed that *pid-2* mutants lost 22G-RNAs from the 5’-end, suggesting a relationship between PID-2 and WAGO-4 activity. Additionally, ZNFX-1 and WAGO-4 form the independent Z-granule [41] where PID-2 is also located. By drawing together the evidence from the results reported here and that available in the literature, it is possible to address the hypothesis that, if the heterodimer has APP-1 activity, PID-5 may interact with PID-2 for the regulated modification of the N-terminus of WAGO-4. This may control WAGO-4 stability and activity and contribute to the balanced inheritance of epigenetic information with ZNFX-1. 

As PID-4/5 may be localised at the periphery of P granules [4], this hypothesis supports the idea that these proteins enable conservation between the perinuclear components and contribute to the spatial–temporal regulation of RNAe components. 

### Therapeutic Translational Research

The potential biological application of the current work is also seen in *Plasmodium falciparum* (*Pf*), a lethal human malaria-causing *Plasmodium* species [8]. Increasing resistance to the antimalarial artemisinin in *Pf* is a significant concern. *Pf*APP-1 has been identified as key to parasitic survival and therefore could be a promising therapeutic target. However, apstatin has been shown to be only a weak inhibitor of *Pf*APP-1, with more potent inhibitors not currently available. However, if dimerisation is shown to be crucial to *Pf*APP-1 activity, a dimerisation block may be of interest therapeutically. Comparative studies such as this contribute to the determination of binding site differences, which is key to designing specific inhibitors to target common catalytic centres/mechanisms.

Inhibitor design is also of interest for HuAPP-1. Studies by Silva et al. [7] demonstrated increased replication-related DNA damage with knocked-down HuAPP-1. The relationship between HuAPP-1 and genome stability suggests that HuAPP-1 inhibitors may be of therapeutic importance for treating malignancies as a lack of HuAPP-1 would result in DNA replication errors, genome instability and ultimately cell death. The narrow specificity of APPs due to the requirement for proline as the second residue makes them an attractive therapeutic target. The specific mechanisms of APP-1 action need to be further studied and enhanced inhibitors can potentially be designed and incorporated into chemotherapeutic or immunotherapy regimes.

## 6. Conclusions

Based on the evidence from the present detailed bioinformatic analysis, we state the following hypotheses: (a) PID-5 APP-RD will share high structural homology with known APP-1 enzymes; (b) PID-5 APP-RD will bind with a single Zn^2+^ atom and is likely to be catalytically inactive and therefore classed as a ‘*pseudoenzyme*’; (c) PID-5 APP-RD will bind apstatin transiently and might also interact with the Pro-rich N-terminal domain of WAGO-4; and (d) PID-5 APP-RD will heterodimerize with *Ce*APP-1 

This study supports the hypothesis that PID-5 APP-RD may function to heterodimerize with *Ce*APP-1. Ultimately, in order to confirm or repute these hypotheses, specific biological experiments must be designed to addressing the above points.

## Figures and Tables

**Figure 1 biomolecules-13-01132-f001:**
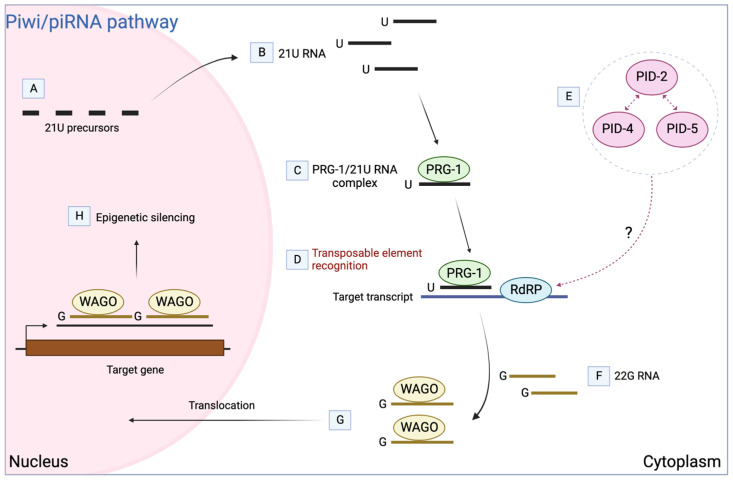
Modular mechanism of the Piwi/piRNA pathway in *C. elegans* (PRG-1/21U RNA) for multigenerational transposable element silencing in the germline. (**A**) Transcription of 21U RNA precursors, small 25–26-nucleotide-long RNAs. (**B**) Truncation of two 5’ precursor bases leave a 5’-uridine and further truncation at 3’-end creates mature 21U-RNA. (**C**) The cognate 21 nucleotide’s long, 5’-uridine, mature 21U RNAs are bound to PRG-1. (**D**) PRG-1/21U RNA complex scans germline transcripts, recognising the target mRNA via imperfect base-pair complementarity and recruits RNA-dependent RNA polymerase (RdRP) to the target. (**E**) PID-4 and PID-5 proteins interact with PID-2 and are hypothesised to play a role in controlling RdPR activity. (**F**) RdRP uses the targeted transcript for synthesis of complementary 22G RNAs that reinforce the PRG-1-initiated silencing mechanism. (**G**) 22G RNAs are loaded onto worm-specific argonaute proteins (WAGOs) and are transported to the nucleus for epigenetic silencing. (**H**) The 22G RNA/WAGO complex initiates silencing through deposition of heterochromatic marks at the target in a way that lasts several generations in a process termed RNA-induced epigenetic gene silencing (RNAe). The way in which the silencing becomes PRG-1-independent is currently unknown. Adapted from Ref. [5].

**Figure 2 biomolecules-13-01132-f002:**
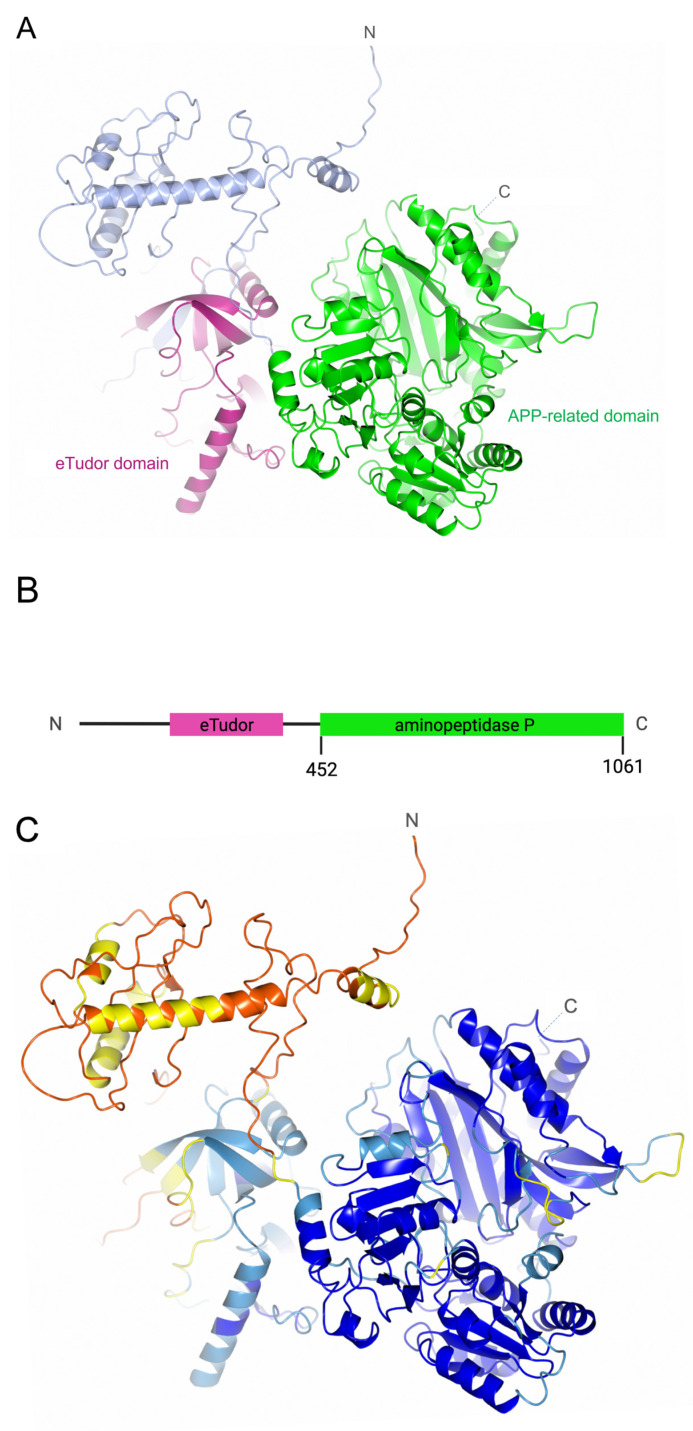
Structure of PID-5 as predicted by AlphaFold2 (AlphaFold2 Database: Q9GUI6). (**A**) The structure of PID-5 with eTudor domain (pink) and APP-RD (green). (**B**) Schematic of the linear PID-5 protein domain composition. The eTudor domain was identified by HHpred (pink) and the APP-RD (green) was identified using BLAST [4]. (**C**) PID-5 coloured by the AlphaFold2 per-residue confidence score (pLDDT). Very high model confidence shown in dark blue (pLDDT > 90), confident in light blue (90 > pLDDT > 70), low confidence in yellow (70 > pLDDT > 50) and very low confidence in orange (pLDDT < 50).

**Figure 3 biomolecules-13-01132-f003:**
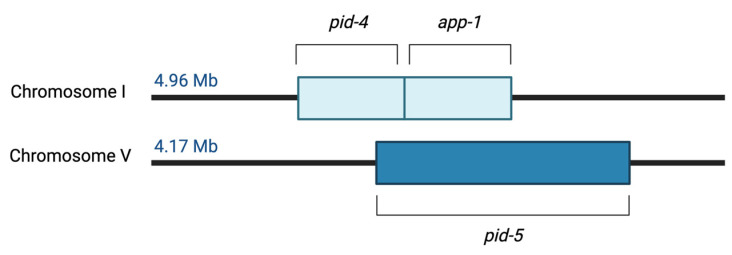
Schematic of the *pid-4, app-1* and *pid-5* genes within the *C. elegans* genome. Genomic data was sourced from EMBL-EBI Ensembl Metazoa Release 56—Feb 2023 (https://metazoa.ensembl.org/index.html) (accessed on 10 April 2023).

**Figure 4 biomolecules-13-01132-f004:**
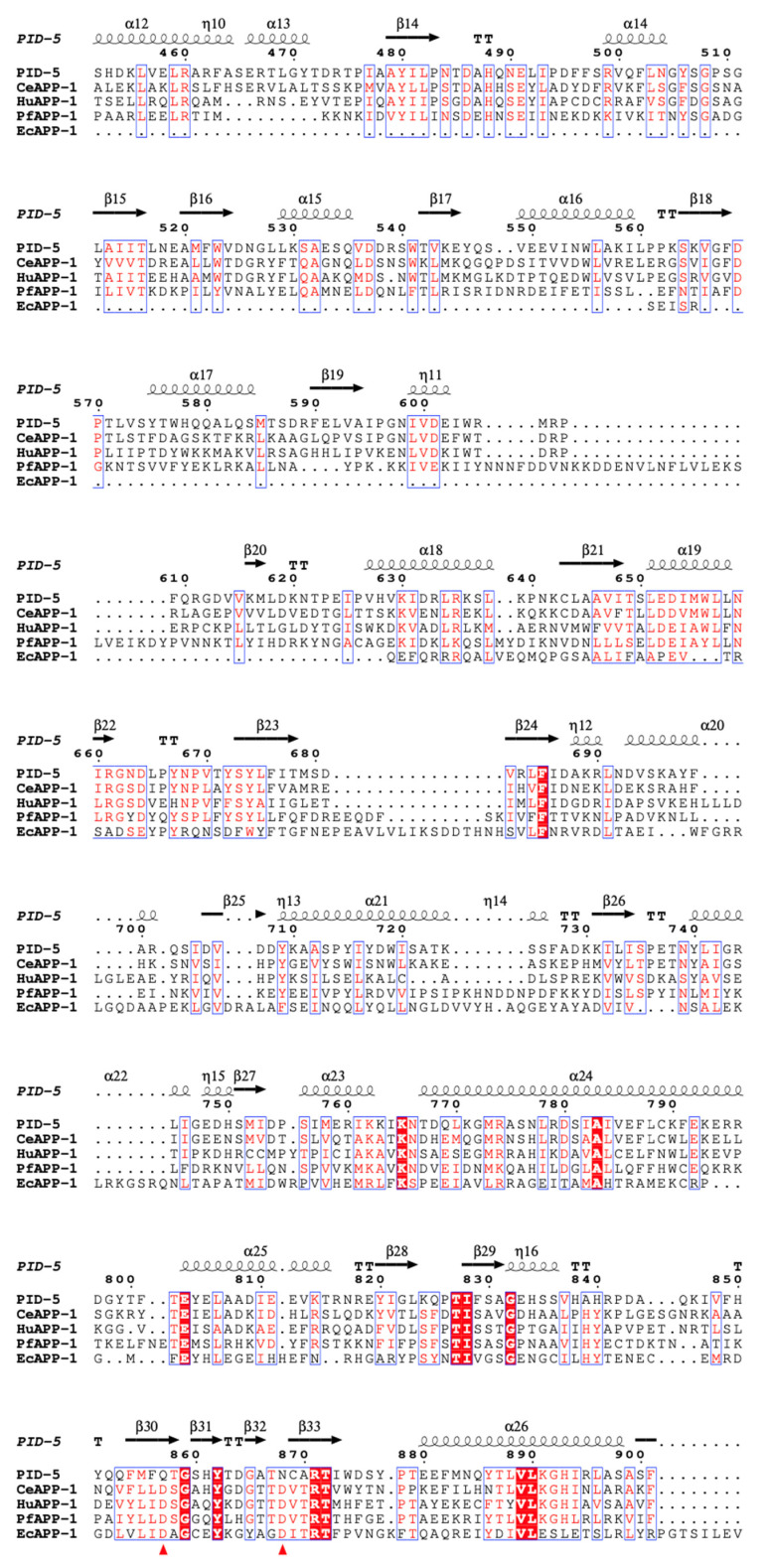
Sequence alignment of PID-5 APP-RD with known homologues from *C. elegans* (*Ce*APP-1), human (HuAPP-1), *P. falciparum* (*Pf*APP-1) and *E. coli* (*Ec*APP-1). Residues denoted in white text with a red background indicate 100% sequence identity. Regions of high sequence similarity are represented by red text and a white background. In the absence of both identity and similarity, residues are shown in black text with a white background. Dots indicate gaps in the protein sequence generated via the alignment process. Secondary structural elements are shown for α-helices (α), β-strands (β) and turns (TT). The secondary structures and residue numbering correspond to PID-5. Residues that coordinate metal ions are highlighted by a red triangle. PDB codes are in parentheses for the following APP homologues: *Ce*APP-1 (PDB code: 4S2R), HuAPP-1 (PDB code: 3CTZ), *Pf*APP-1 (PDB code: 5JQK) and *Ec*APP-1 (PDB code: 1WL9). Only the APP-RD of PID-5 was included in the alignment (S452-I1061) (AlphaFold2 DB: Q9GUI6). Figure produced using ESPript 3.0 (https://espript.ibcp.fr/ESPript/ESPript/ (accessed on 28 February 2023)) [21].

**Figure 5 biomolecules-13-01132-f005:**
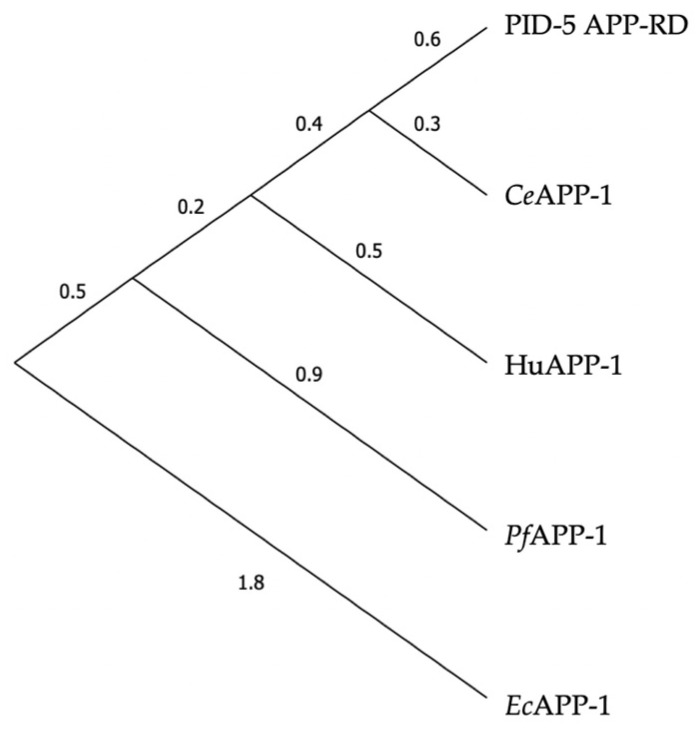
Phylogenetic tree of APP-1 structures and PID-5 APP-RD. Maximum likelihood tree for *Ce*APP-1 (PDB code: 4S2R), HuAPP-1 (PDB code: 3CTZ), *Pf*APP-1 (PDB code: 5JQK), *Ec*APP-1 (PDB code: 1WL9) and PID-5 APP-RD (AlphaFold2 DB: Q9GUI6, residues S452-I1061 only).

**Figure 6 biomolecules-13-01132-f006:**
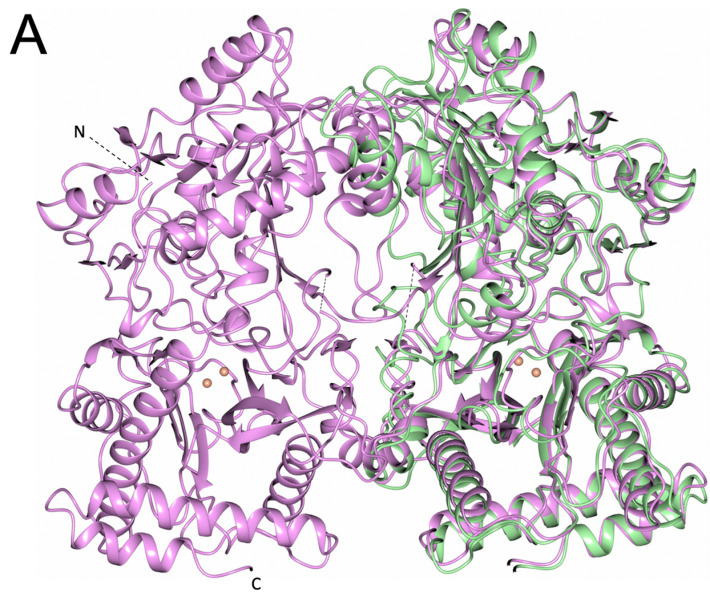
3D Comparison of PID-5 APP-RD with *Ce*APP-1. (**A**) The *Ce*APP-1 homodimer is shown in pink (PDB code: 4S2R) with PID-5 APP-RD shown in green (AlphaFold2 DB: Q9GUI6, Ser452-Ile1061). The structures superimpose an RMSD of 1.8 Å over 601 aligned Cα atoms. The divalent zinc ions from *Ce*APP-1 are shown in pale pink. Black dashed lines show the flexible regions of *Ce*APP-1. (**B**) C-terminal domain canonical ‘*pita-bread fold*’ of the clan MG peptidases. (**a**) *Ce*APP-1 and (**b**) PID-5 APP-RD. α-helices are shown in purple and β-strands in blue.

**Figure 7 biomolecules-13-01132-f007:**
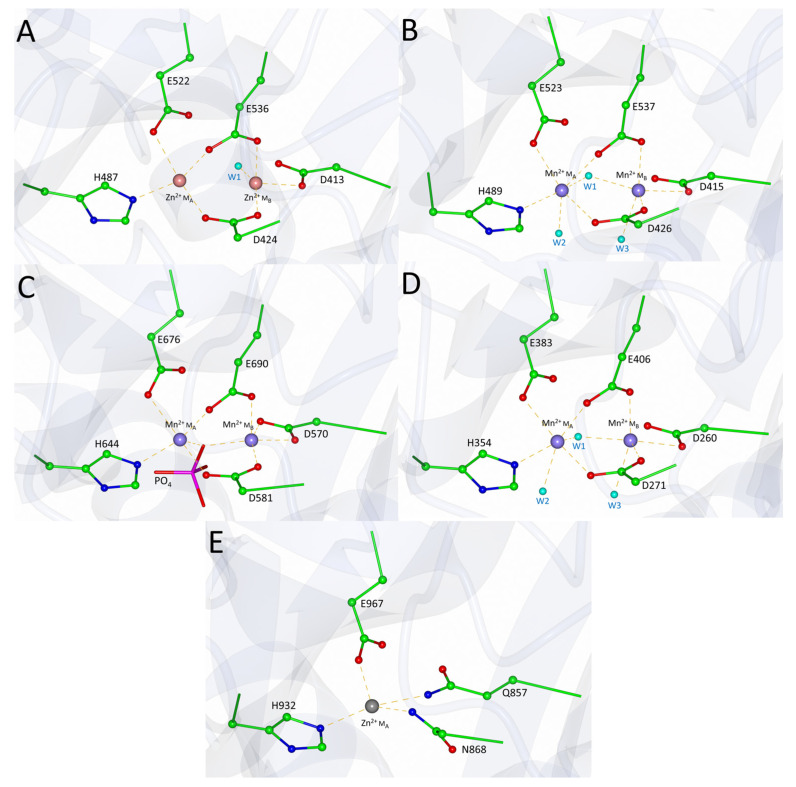
Metal coordination site geometry of APP-1 homologues and PID-5 APP-RD. Orientation of the metal-coordinating residue side chains of the homologous structures (**A**–**D**) and PID-5 APP-RD predicted structure (**E**) (AlphaFold2 DB: Q9GUI6). Pink spheres represent the coordinated divalent zinc metal ions and purple spheres represent manganese. Water molecules are shown as smaller, light blue spheres. Yellow dashed lines show the metal coordination interactions. PDB codes in parentheses for the analysed structures: (**A**) *Ce*APP-1 (PDB code: 4S2R), (**B**) HuAPP-1 (PDB code: 3CTZ), (**C**) *Pf*APP-1 (PDB code: 5JQK) and (**D**) *Ec*APP-1 (PDB code: 1WL9).

**Figure 8 biomolecules-13-01132-f008:**
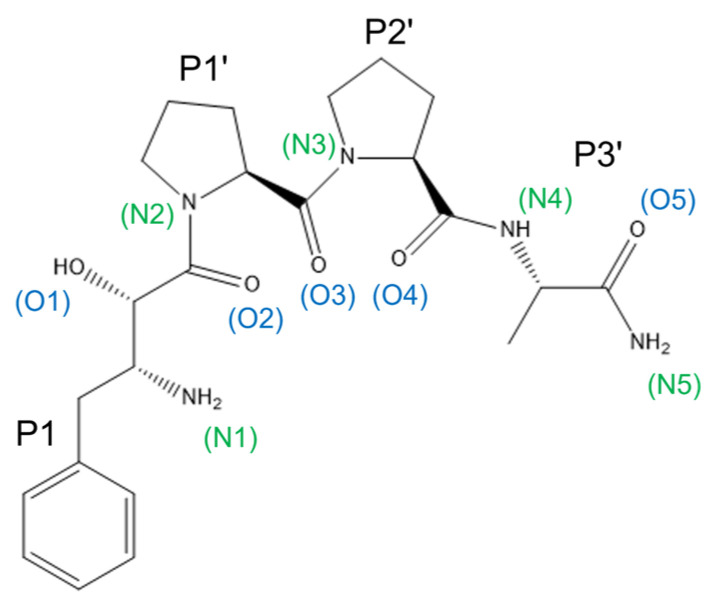
The structure of APP-1 inhibitor apstatin. Apstatin is annotated with its own naming convention of the oxygen and nitrogen atoms for clarity in analysis. The hydroxyl group of the N-terminal (2S,3R)- 3-amino-2-hydroxy-4-phenyl-butanoic acid (O1) binds both metal ions [8]. The two proline residues accommodate the enzyme’s subsite specificity for the P1’ Pro and P2’ Pro and the amino acid amide Ala-NH_2_ was designed based on the higher binding affinities of tetrapeptides [33]. Figure created in Chem3D.

**Figure 9 biomolecules-13-01132-f009:**
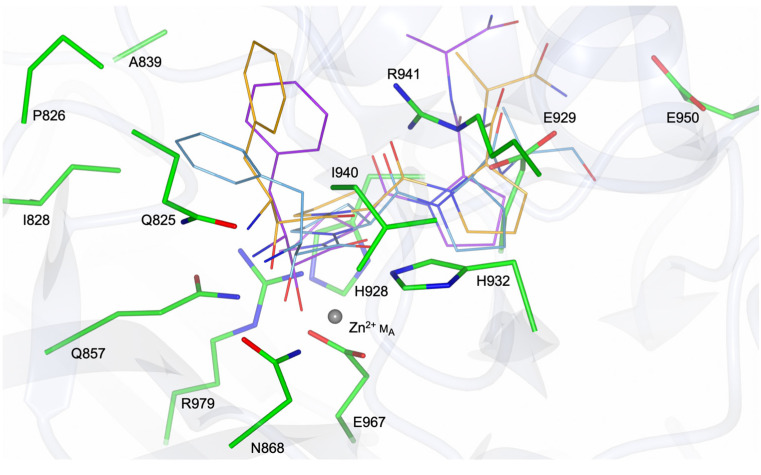
Putative PID-5 APP-RD binding site residues with apstatin orientations from the homologous APP-1 crystal structures. The amino acid side chains of PID-5 APP-RD, which may contribute to apstatin binding, are shown (green) (AlphaFold2 DB: Q9GUI6). The active site residues were identified from superimposition with known APP-1:apstatin crystal structures. The blue, orange and purple apstatin orientations are taken from *Ce*APP-1, *Pf*APP-1 and *Ec*APP-1, respectively. The grey sphere represents the predicted zinc ion in PID-5 APP-RD. PDB codes in parentheses for analysed structures in complex with apstatin: *Ce*APP-1 (PDB code: 4S2T), *Pf*APP-1 (PDB code: 5JR6) and *Ec*APP-1 (PDB code: 1N51).

**Figure 10 biomolecules-13-01132-f010:**
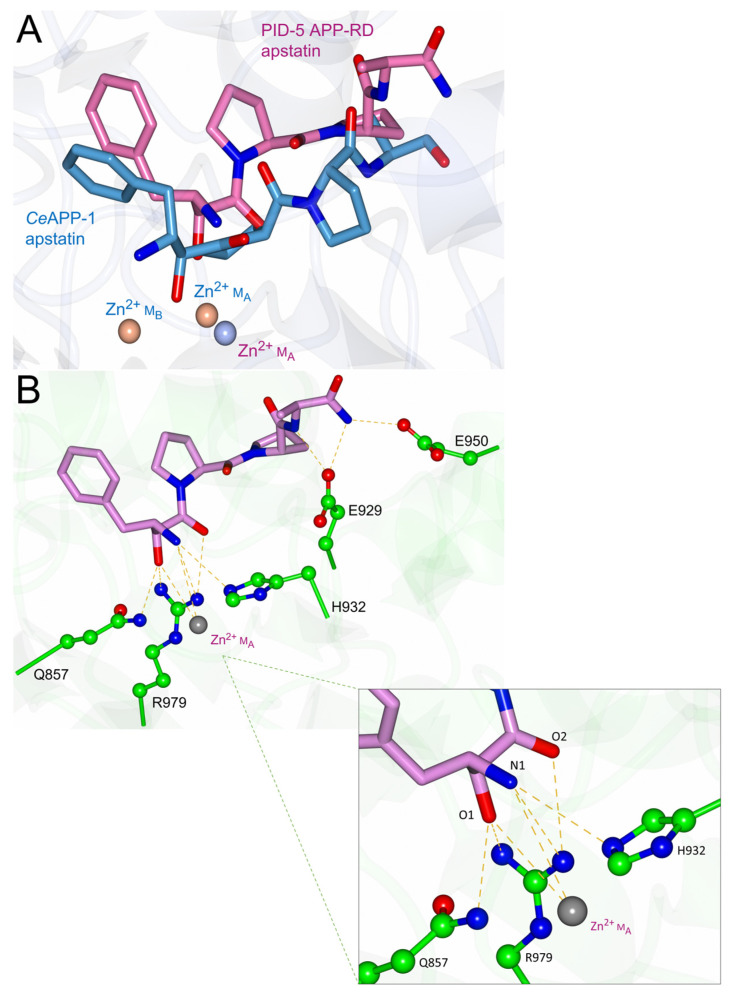
Docking of PID-5 APP-RD predicted structure with apstatin. (**A**) Orientation of the highest ChemScore docking solution (pink), in comparison to the apstatin position from *Ce*APP-1 (PDB code: 4S2T) (blue). Light pink spheres show the Zn^2+^ ions from *Ce*APP-1 and the grey sphere represents the predicted zinc in PID-5 APP-RD. (**B**) PID-5 APP-RD:apstatin interactions (AlphaFold2 DB: Q9GUI6). The side chains of residues that could potentially form hydrogen-bonding interactions with apstatin are shown, with predicted bonds shown by yellow dashes.

**Figure 11 biomolecules-13-01132-f011:**
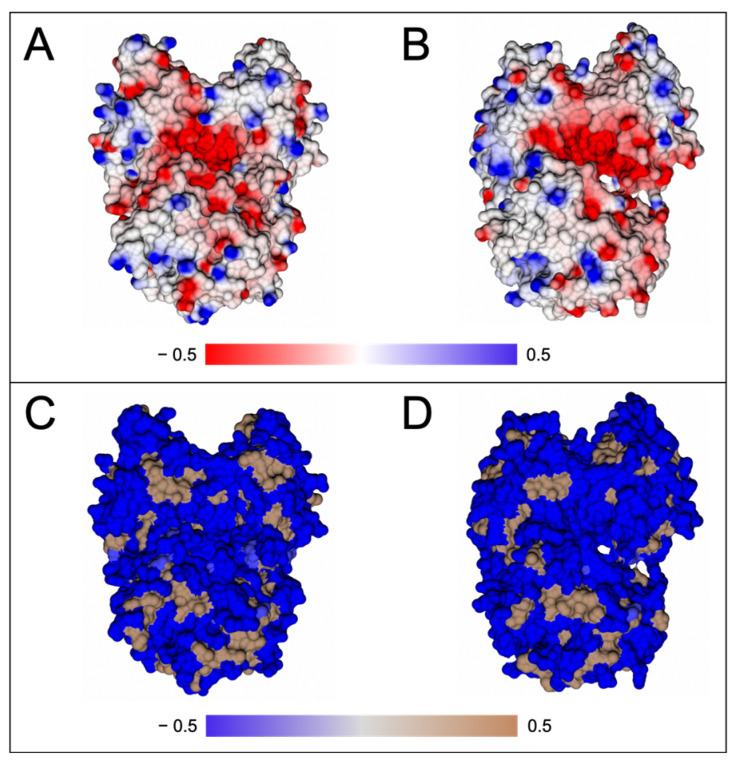
The charge and hydrophobicity landscape at the cross section of the dimer interface. Charges were mapped onto the molecular surface using the Poisson–Boltzmann solver within CCP4mg for (**A**) *Ce*APP-1 (PDB code: 4S2R) and (**B**) PID-5 APP-RD (AlphaFold2 DB: Q9GUI6, S452-I1061). Red areas represent negatively charged patches, blue areas are positively charged and areas in white represent close-to-neutral residues. The hydrophobicity of the surface is shown for (**C**) *Ce*APP-1 and (**D**) PID-5 APP-RD. Pale brown areas highlight hydrophobic patches as generated by the GRID-type hydrophobic potential in CCP4mg.

**Figure 12 biomolecules-13-01132-f012:**
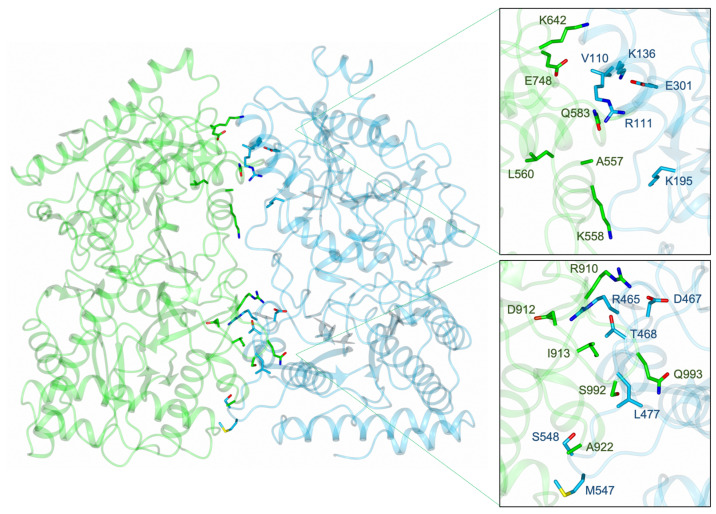
Predicted PID-5 APP-RD and *Ce*APP-1 heterodimer. The heterodimer of PID-5 APP-RD (AlphaFold2 DB: Q9GUI6, residues S452-I1061) and *Ce*APP-1 (PDB code: 4S2R) was predicted by AlphaFold2. The side chains of the interface residues are shown for both PID-5 APP-RD (green) and *Ce*APP-1 (blue).

**Figure 13 biomolecules-13-01132-f013:**
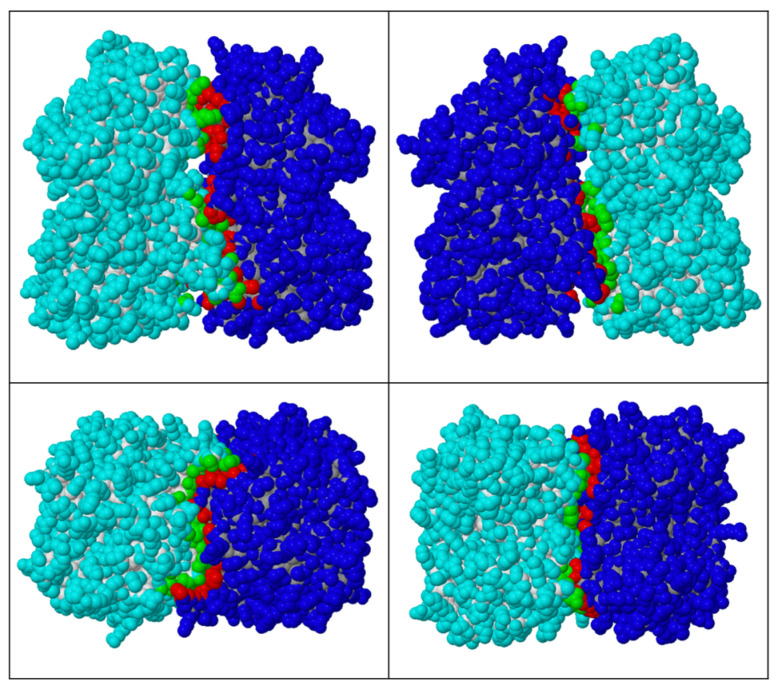
Space-filling representation of the predicted PID-5 APP-RD and *Ce*APP-1 heterodimer. The heterodimer of PID-5 APP-RD (AlphaFold2 DB: Q9GUI6, S452-I1061) and *Ce*APP-1 (PDB code: 4S2R) was predicted by AlphaFold2. The spacefill rendering was performed by PDBePISA, where the *Ce*APP-1 monomer is shown in dark blue, with dimer interface residues in red, PID-5 APP-RD in light blue, and dimer interface residues in green.

**Table 1 biomolecules-13-01132-t001:** Percentage sequence identity and similarity of known APP-1 structures to PID-5 APP-RD using EBLOSUM62 matrix. Pairwise sequence alignment was performed through the EMBL-EBI EMBOSS Needle tool. The associated PDB codes are shown for the homologous APP-1 structures. Identity shows percentage of perfectly conserved amino acid residues and similarity shows percentage of amino acids with similar physiochemical properties.

	PDB Code	Identity (%)	Similarity (%)
*Ce*APP-1	4S2R	41.4	61.4
HuAPP-1	3CTZ	33.7	51.7
*Pf*APP-1	5JQK	25.0	45.7
*Ec*APP-1	1WL9	16.6	27.8

**Table 2 biomolecules-13-01132-t002:** Pairwise comparison of PID-5 APP-RD to *Ce*APP-1, HuAPP1, *Pf*APP-1, and *Ec*APP-1 using DALI. PID-5 APP-RD (AlphaFold2 DB: Q9GUI6, residues Ser452-Ile1061) was used as the query structure and compared against the known APP-1 structures (one-against-many comparison). Structures in complexes with the APP-1 inhibitor apstatin are also shown.

Rank	Protein	Z-Score	RMSD (Å)	ID (%)	Chain
1	*Ce*APP-1	47.9	1.8	44	4S2R-Q
2	*Ce*APP-1:Apstatin	47.9	1.7	44	4S2T-Q
3	HuAPP-1	43.3	2.1	34	3CTZ-A
4	*Pf*APP-1:Apstatin	41.9	2.0	26	5JR6-A
5	*Pf*APP-1	41.8	1.8	26	5JQK-A
6	*Ec*APP-1:Apstatin	21.0	3.2	18	1N51-A
7	*Ec*APP-1	20.9	3.4	16	1WL9-A

**Table 3 biomolecules-13-01132-t003:** Metal-coordinating residues of APP-1 homologues and PID-5 APP-RD. The conserved metal-coordinating residues from the homologous structures are separated into the two distinct metal binding sites. The equivalent residues in the PID-5 APP-RD predicted structure (AlphaFold2 DB: Q9GUI6) are aligned. Residues shown in green text are conserved across all five structures. Residues shown in blue are conserved across the homologues only. Black text highlights the residues which are not conserved. PDB codes in parentheses: *Ce*APP-1 (4S2R), HuAPP-1 (3CTZ), *Pf*APP-1 (5JQK) and *Ec*APP-1 (1WL9).

	PID-5 APP-RD	*Ce*APP-1	HuAPP-1	*Pf*APP-1	*Ec*APP-1
M_A_	N868	D424	D426	D581	D271
	E967	E522	E523	E676	E383
	G981	E536	E537	E690	E406
	H932	H487	H489	H644	H354
M_B_	Q857	D413	D415	D570	D260
	N868	D424	D426	D581	D271
	G981	E536	E537	E690	E406

**Table 4 biomolecules-13-01132-t004:** AlphaFill small molecule transplant results. The ‘transplanted’ compound is shown with percentage sequence identity between PID-5 APP-RD and the reference PDB-REDO entry and its PDB code. The global root-mean-square deviation (RMSD) values between the structurally aligned PID-5 APP-RD of the AlphaFold2-predicted structure (AlphaFold2 DB: Q9GUI6, Ser452-Ile1061), and the donor structure Cα-atoms are shown. Local RMSD (local structural alignment of backbone atoms within 6 Å) and Transplant Clash Score (TCS) (that represent the van der Waals overlap between the inserted ions and the protein binding site within 4 Å) are shown as quality indicators [14]. Scores highlighted in red represent medium-confidence results.

Compound	Identity (%)	PDB.Chain	Global RMSD (Å)	Local RMSD (Å)	TCS
Zn	40	4S2R.A	2.12	1.13	0.16
Mn	30	3CTZ.A	3.66	0.33	0.19
	25	5CDL.A	2.97	0.39	0.00
Ca	30	3CTZ.A	3.66	1.36	0.05
	25	3Q6D.A	2.10	0.34	0.47
Na	30	3CTZ.A	3.66	0.37	0.11
	25	5CDV.A	2.98	0.41	0.00
	25	2ZSG.A	9.33	0.02	0.28
	25	5GIU.A	2.98	0.37	0.06
	25	2ZSG.B	9.81	2.34	0.89
Co	25	1WN1.A	2.55	0.36	0.28
	25	1WN1.A	2.55	0.42	0.00

**Table 5 biomolecules-13-01132-t005:** Apstatin binding site residues of known APP-1 structures and the predicted PID-5 APP-RD structure. Table of identified apstatin-interacting residues for *Ce*APP-1 (PDB code: 4S2T), *Pf*APP-1 (PDB code: 5JR6) and *Ec*APP-1 (PDB code: 1N51), with equivalent residues aligned and corresponding residues in PID-5 APP-RD identified (AlphaFold2 DB: Q9GUI6, S452-I1061). Apstatin from the inhibitor-bound *Ce*APP-1 structure was modelled into HuAPP-1 in the absence of an apstatin-bound crystal structure. Residues in green text are conserved across all structures and residues in blue are conserved across three structures. Residues in black text are not conserved across the structures. Residues listed in bold text have been specifically referenced in the published literature. Interactions detailed for *Ce*APP-1:apstatin were used to map many of the other structure’s residues as a discrepancy in the level of detail was observed in the literature. Those residues mapped from structural superimposition of another structure are shown in normal print.

PID-5 APP-RD	*Ce*APP-1	*Pf*APP-1	*Ec*APP-1	HuAPP-1
L492	**Y43**	**I163**	-	Y42
D525	**D76**	N196	-	D75
G527	**R78**	L198	-	R77
Q825	**F378**	**F537**	**Y229**	F381
P826	**D379**	S538	N230	P382
I828	**I381**	I540	I232	I384
A839	**H392**	**H551**	**H243**	H395
Q857	**D413**	**D570**	**D260**	D415
N868	**D424**	**D581**	**D271**	D426
H928	**H483**	**H640**	**H350**	H485
E929	**G484**	G641	G351	G486
G931	**G486**	G643	S353	G488
H932	**H487**	**H644**	**H354**	H489
I940	V495	**V652**	**V360**	V497
R941	**H496**	**H653**	**H361**	H498
E950	**R505**	V662	**R370**	K507
E967	**E522**	**E676**	**E383**	E523
R979	**R534**	R688	**R404**	R535
G981	**E536**	**E690**	**E406**	E537

**Table 6 biomolecules-13-01132-t006:** Conserved dimer interface residues. The dimer interface residues for *Ce*APP-1 (PDB code: 4S2R) with the equivalent residues in PID-5 APP-RD (AlphaFold2 DB: Q9GUI6) aligned. Hydrogen-bonding interactions and salt bridges between *Ce*APP-1 Chain Q and P are listed. The equivalent potential interactions in the modelled PID-5 APP-RD homodimer are shown in the latter two columns. Potential heterodimer interactions can be inferred through the central two columns and the outer two columns. Residues in green text are conserved in both *Ce*APP-1 and PID-5 APP-RD and blue text highlights similar residues. Backbone interactions are indicated by an asterix.

		*Ce*APP-1 Chain Q	*Ce*APP-1 Chain P	PID-5 APP-RD (Chain Q Equivalent)	PID-5 APP-RD (Chain P Equivalent)
Hydrogen bonds	1	K195	V110 *	K642	A557
	2	K195	R111 *	K642	K558
	3	K195	L113 *	K642	L560
	4	K136	E301	Q583	E748
	5	T468	R465 *	I913	R910
	6	M547 *	L477 *	S992	A922
	7	S548 *	L477 *	Q993	A922
	8	S548	G478 *	Q993	G923
	9	V110 *	K195	A557	K642
	10	R111 *	K195	K558	K642
	11	E301	K136	E748	Q583
	12	R465 *	T468	R910	I913
	13	D467	R465	D912	R910
	14	L477 *	M547 *	A922	S992
	15	L477 *	S548 *	A922	Q993
	16	G478 *	S548	G923	Q993
Salt bridges	1	K136	E301	Q583	E748
	2	R465	D467	R910	D912
	3	E301	K136	E748	Q583
	4	D467	R465	D912	R910

**Table 7 biomolecules-13-01132-t007:** PDBePISA dimer interface analysis for *Ce*APP-1 homodimer (PDB code: 4S2R), and the AlphaFold2-predicted PID-5 APP-RD homodimer and heterodimer with *Ce*APP-1. The predicted PID-5 APP-RD was used in the dimer models (AlphaFold2 DB: Q9GUI6, residues S452-I1061). The interface area is also shown as a percentage of the total solvent-accessible area buried upon dimer formation.

Protein Dimer	Solvent-Accessible Interface Area	
	(Å^2^)	(%)
*Ce*APP-1: *Ce*APP-1	2005.3	7.7
*Ce*APP-1: PID-5 APP-RD	1782.6	6.7
PID-5 APP-RD: PID-5 APP-RD	1574.4	5.8

## Data Availability

No new data created.

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
