# Peer review of "A Molecular Analysis of the Aminopeptidase P-Related Domain of PID-5 from Caenorhabditis elegans"

_biomolecules, 2023, doi:10.3390/biom13071132_

Round 1

Reviewer 1 Report

The manuscript presented by Lloyd and coworkers entitled "A molecular analysis of aminopeptidase P-related domain of PID-5 from Caenorhabditis elegans" is sound, credible and clearly advances current understanding in the interactions formed by PID-5 APP-RD. The experimental evidence available is well-coupled with in silico studies to present a thorough analysis of the macromolecular target, including its role as a metal binding protein. The importance of Structural Bioinformatics making the most of the latest advances of AlphaFold and AlphaFill is highlighted through the well-justified prediction of potential interactions.  

Author Response

No change suggested by the Reviewer.

Reviewer 2 Report

The manuscript by Lloyd et al. describes a detailed analysis of an AlphaFold2-predicted model of the aminopeptidase P-related domain of PID-5 from Caenorhabditis elegans. The authors suggest that, differently to other APPs, this domain may contain only a single metal site, that it may be Zn2+ and that the enzyme may be inactive. They show that it may bind the APP inhibitor apstatin albeit weakly and that it can form heterodimers with PID-2.

The work is well-executed and mostly well-presented and thus worthy of publication. Though no original experimental work is presented, the in silico results present interesting hypotheses that can stimulate future experiments. 

In the Introduction, I felt that "The maternal piRNAs" mentioned on line 62 were not sufficiently well-connected to the mechanism in Figure 1.

I am not an expert on RNAe, but it seems that the statement on lines 52-53 is contradictory. How can a protein silence its target mRNA at the same time as it recruits an RNA polymerase?

I think that an additional figure of the AlphaFold2 model coloured by reliability score should be included, perhaps as a supplementary figure, but preferably as an additional panel in Figure 2.

On line 137, what algorithm was used to superpose the structures in Coot? I assume SSM. If so, give the reference.

On line 183, the reader will not know where the residues with low confidence are. Can they be indicated on a figure?

Line 205: The MSA helps asses the evolutionary relationships between the sequences, not (directly) the structures. Sometimes the structures are more similar than sequence similarity suggests.

Figure 6A: When looking at small structural differences I think it would help to use thinner helices so that the differences are not obscured by their thickness. Also, more contrasting colours for the two structures would help. Both are a bit dark as is.

Line 291: Connect the PID-5 structure to panel E.

Line 313 and onwards: Connect the structures you mention to their PDB IDs in the Table 4.

Figure 9 is a bit cluttered. Would it be possible to give it more of a 3-dimensional feeling? Perhaps some depth fog?

Line 451: Why was it necessary to superpose PID-5 APP-RD on both CeAPP-1 chains to assess heterodimerisation? Wouldn't one have been enough? Or was it a cross-check?

I feel that the discussion of interface residues and their conservation would be much enhanced by a figure.

Line 528: Which enzyme is made inactive, the human or E. coli one?

Line 537: Clarify what apstatin is binding to in this sentence.

Small grammatical suggestions:

line 52: "scan" should be "scans"

line 58: Add "The" at the beginning of the sentence

line 92: remove "the" before "PID-5"

line 147: I assume it wasn't the structure of apstatin that was generated in Chem3D but rather coordinates and a geometry description?

line 171: "inform on the role"

line 185: should be "an MSA". Also, define the abbreviation. I don't think this has been done earlier in the manuscript. 

line 248: "APP's" should be "APPs"

line 355: I would write "designed based on" rather than "designed due to"

line 443: Shouldn't this be "ΔGi"?

line 558: "shown to be"

Author Response

Line 62: Changed to “Maternally provided”.

Lines 52-53: Agree - removed “and silencing” for clarity.

Alphafold2 PID-5 figure with confidence scores included as Figure 2C.

Line 137: Added in SSM and repeated the Coot reference.

Line 183: Referenced Figure 2C.

Line 205: Changed “structures” to “sequences”.

Figure 6A: Helices made thinner and lighter, more contrasting colours were chosen.

Line 291: Added “(E)”.

Line 313: PDB codes added for reference structures throughout paragraph.

Figure 9: Added depth fog, reduced cylinder size and reduced font size.

Line 451: Yes, this was for cross check and the analysis focussed on one region. This has been reworded for clarity.  

Interface residue figure: New Figure (12) added showing the AlphaFold2 predicted heterodimer and the side chains of the residues on the interface.  

Line 528: Clarified that it is E. coli and specified the residues.

Line 537: Added “to PID-5 APP-RD”.

Language:

Line 52: “scan” changed to “scans”.

Line 58: Added “the” at beginning of sentence.

Line 92: Removed “the” before PID-5.

Line 147: Changed to coordinates and geometry.

Line 171: Added “on”.

Line 185: MSA abbreviation defined on line 180, “an” added.

Line 248: Changed to “apps”.

Line 355: Changed to ‘designed based on’.

Line 443: “i” changed to superscript as written on the PDBePISA interface.

Line 558: Changed to “shown to be”.

Reviewer 3 Report

Lloyd et al. have compared the predicted three-dimensional structure of PID-5 with aminopeptidase P molecular structure. Structural homology, metal binding site, apstatin inhibitor binding site and dimerization interface of these protein are compared to decipher the catalytic and biochemical mode of function of PID-5. The manuscript has many strengths and is a nice piece of in silico work which can be helpful in future experimental work. Overall, the manuscript is well written, in silico experiments are well performed and discussed. I have the following concerns about the manuscript.

1.       The full form of PID-5 should be mentioned at first instance.

2.       The structure of the Oligomer (dimer) of PID-5 should be predicted by alpha fold. Moreover, the heterodimerization speculated in the section “PID-5 APP-RD could heterodimerize with CeAPP-1” should be tested by the prediction of heterodimer structure by AlphaFold.

3.       Furthermore, using the predicted structure of heterodimer, the dimerization strength should be calculated by PISA server.

4.       Figure 12: It is unclear why some atoms are shown in red.

5.       Figure 11: The scale of charge and hydrophobicity is missing. It should be included. How are the charge and hydrophobicity surface calculated? – should be clarified.

6.       Figure 7 and 10: The distances for all interaction shown by yellow-dashed line should be mentioned.

7.       Figure 6: The R.M.S.D value for superimposition of two structures should be provided.

8.       Figure 5: The scale for branch length should be provided.

Author Response

  1. Added ‘from a “piRNA-induced silencing defective” (Pid) mutation screen’ to highlight the origin of the PID-5 name.
  2. AlphaFold2 dimer prediction – Performed using the ColabFold software and has been added to the methods and the reference. The highest ranked structure prediction for both the PID-5 APP-RD homodimer and heterodimer with CeAPP-1, were as we had predicted / modelled in our analysis.

Links for reference:

https://github.com/sokrypton/ColabFold

https://www.nature.com/articles/s41592-022-01488-1

  1. Dimerization strength calculated by PDBePISA – Table 7 has been updated with the PDBePISA results for the AlphaFold2 predicted structures. Figure 12 has also been updated to be a space-filling representation of the AlphaFold2-predicted heterodimer.
  2. Added “with dimer interface residues in red” in the Figure 12 legend
  3. Figure 11: How the charge and hydrophobicity was calculated is included in the methods section 2.4. ‘The protein surface charge potential and hydrophobicity was calculated using CCP4mg’. Charge and hydrophobicity scale now included using the CCP4mg defined parameters.
  4. Distances for all interactions with yellow-dashed line (Figure 7 and 10). We do not think it is necessary to include the distances of the interactions in Figure 7A-D as the figures will become cluttered and the focus is on residue conservation rather than distances. Figure 7E and 10 are predictions and therefore we would argue the precise distances are not required on the figure.
  5. R.M.S.D Figure 6 – Added in “The structures superimpose with an RMSD of 1.8 Å over 601 aligned Cα atoms”.
  6. Figure 5 New phylogenetic tree created with MEGA including branch lengths and methods updated with these details and reference.